# Autism Spectrum Disorder: A Neuro-Immunometabolic Hypothesis of the Developmental Origins

**DOI:** 10.3390/biology12070914

**Published:** 2023-06-26

**Authors:** Martin G. Frasch, Byung-Jun Yoon, Dario Lucas Helbing, Gal Snir, Marta C. Antonelli, Reinhard Bauer

**Affiliations:** 1Department of Obstetrics and Gynecology, University of Washington, Seattle, WA 98195, USA; snirg@uw.edu; 2Center on Human Development and Disability, University of Washington, Seattle, WA 98195, USA; 3Electrical and Computer Engineering, Texas A&M University, College Station, TX 77843, USA; bjyoon@tamu.edu; 4Institute for Molecular Cell Biology, Jena University Hospital, Friedrich Schiller University, 07747 Jena, Germany; dario-lucas.helbing@med.uni-jena.de (D.L.H.); reinhard.bauer@med.uni-jena.de (R.B.); 5Leibniz Institute on Aging, Fritz Lipmann Institute, 07745 Jena, Germany; 6Department of Psychiatry and Psychotherapy, Jena University Hospital, Friedrich Schiller University Jena, 07747 Jena, Germany; 7Center for Intervention and Research on Adaptive and Maladaptive Brain Circuits Underlying Mental Health (C-I-R-C), Jena-Magdeburg-Halle, 07743 Jena, Germany; 8Instituto de Biología Celular y Neurociencia “Prof. Eduardo De Robertis”, Facultad de Medicina, Universidad de Buenos Aires, Buenos Aires 1121, Argentina; mca@fmed.uba.ar; 9Institute for Advanced Study, Technical University of Munich, Lichtenbergstrasse 2 a, 85748 Garching, Germany

**Keywords:** autism spectrum disorder, information theory, genomics, multi-scale modeling

## Abstract

**Simple Summary:**

According to CDC, autism spectrum disorder (ASD) was diagnosed in one of every 36 children in the United States in 2020. Over 2000 genes have been identified as altered in ASD, yet a coherent model of ASD etiology is lacking. Here, we present a novel hypothesis of ASD origins based on common gene network features involving an interplay of glial and neuronal cells in the developing brain, starting during fetal development. We report statistical findings with implications for understanding the causes of ASD, early detection, and the development of new treatments.

**Abstract:**

Fetal neuroinflammation and prenatal stress (PS) may contribute to lifelong neurological disabilities. Astrocytes and microglia, among the brain’s non-neuronal “glia” cell populations, play a pivotal role in neurodevelopment and predisposition to and initiation of disease throughout lifespan. One of the most common neurodevelopmental disorders manifesting between 1–4 years of age is the autism spectrum disorder (ASD). A pathological glial–neuronal interplay is thought to increase the risk for clinical manifestation of ASD in at-risk children, but the mechanisms remain poorly understood, and integrative, multi-scale models are needed. We propose a model that integrates the data across the scales of physiological organization, from genome to phenotype, and provides a foundation to explain the disparate findings on the genomic level. We hypothesize that via gene–environment interactions, fetal neuroinflammation and PS may reprogram glial immunometabolic phenotypes that impact neurodevelopment and neurobehavior. Drawing on genomic data from the recently published series of ovine and rodent glial transcriptome analyses with fetuses exposed to neuroinflammation or PS, we conducted an analysis on the Simons Foundation Autism Research Initiative (SFARI) Gene database. We confirmed 21 gene hits. Using unsupervised statistical network analysis, we then identified six clusters of probable protein–protein interactions mapping onto the immunometabolic and stress response networks and epigenetic memory. These findings support our hypothesis. We discuss the implications for ASD etiology, early detection, and novel therapeutic approaches. We conclude with delineation of the next steps to verify our model on the individual gene level in an assumption-free manner. The proposed model is of interest for the multidisciplinary community of stakeholders engaged in ASD research, the development of novel pharmacological and non-pharmacological treatments, early prevention, and detection as well as for policy makers.

## 1. Introduction

A body of preclinical and epidemiological evidence shows that in utero exposure to infection or stress results in altered neurodevelopmental trajectories, including psychiatric conditions such as autism spectrum disorder (ASD) [1].

However, the mechanisms explaining the individual risk and framework for early detection and treatment have remained elusive.

ASD is a complex neurodevelopmental disorder that is thought to be caused by a combination of genetic and environmental factors. ASD may be caused by changes in genes that are involved in brain development and function. Environmental factors that may contribute to the development of ASD include exposure to toxins, infections, medications such as valproic acid and chronic stress experienced during pregnancy.

There is no single test that can diagnose ASD. Doctors typically diagnose ASD based on a child’s developmental history, behavior, and medical exam. There is no cure for ASD, but there are treatments that can help improve symptoms and quality of life. Treatment for ASD may include early intervention services, speech therapy, occupational therapy, and medication.

It is important to note that not everyone who is exposed to these risk factors will develop ASD. ASD is a complex disorder that is caused by a combination of factors.

Here, we synthesize the evidence of connections between fetal neuroinflammation, prenatal stress (PS) and ASD to generate a new model of ASD etiology accounting for the contribution of microglia and astrocytes, the key players in brain inflammation. We deployed a multi-species approach to data synthesis [2]. Combining biomarkers derived from observations in multiple species, such as rodent, ovine and human in the present case, should increase the likelihood of identifying a translationally valid disease etiology.

As a result, we propose a novel model of neuro-immunometabolic etiology of ASD. By way of an initial validation of this model on the population level, in a large genome database sample of children affected by ASD, we tested for the presence of disruptive mutations to genes involved in the respective pathways and systems. We conclude with discussion of the implications and recommendations for assumption-free validation of the proposed model in the existing genomic data on the individual level, the subject of an ongoing research study. The proposed model is of interest for the multidisciplinary community of stakeholders engaged in ASD research, the development of novel pharmacological and non-pharmacological treatments, early prevention, and detection as well as for policy makers.

## 2. Methods

### Gathering the Biomarkers across Cell Systems and Species

Based on our observations in fetal microglia [3,4], the simplistic hypothesis is that exposure to endotoxins or chronic stress augments energy conservation-driving pathways, and the α7 nicotinic acetylcholine receptor α7nAChR agonism reduces that response. Such gene-environment adaptations should be reflected in changes of epigenetic memory mediated by histone acetylation/deacetylation enzymes (HDAC/HAT) involved in microglial memory of inflammation [3].

Synthesizing findings from the studies in the rodent, ovine and human species, we identified genes associated with the inflammatory, stress and metabolic phenotypes of microglia and astrocytes (Figure 1 and Table 1). We included the genes involved in energy homeostasis based on observations linking neuroimmune and metabolic memory signatures in fetal microglia exposed to LPS and the larger emerging gestalt of physiological stress response in immunometabolism, in particular by astrocytes [5,6].

We included the iron homeostasis genes based on the recently postulated interaction between this signaling network and the α7nAChR modulation of the immunometabolic phenotype of fetal microglia [4].

We included the complement pathway genes expressed in microglia and known to be modulated by neuroinflammation in microglia and to play an important role in neurodevelopmental synaptic pruning [4,7].

Based on the evidence that in utero inflammation or stress increases risk for ASD, we sought to identify genes linked to the ASD diagnosis and falling into the family of signaling pathways listed in Table 1. We tested that hypothesis statistically against the Simons Foundation Autism Research Initiative (SFARI) Gene database. The SFARI Gene database is an online resource for genes implicated in autism susceptibility. The SFARI gene database is a continually updated repository of genetic risk factors for ASD as they emerge from research. We used the human gene module, which is freely accessible online and contains 1231 genes identified as playing a role in ASD. This module is combined with a gene scoring module that assesses quantitatively the levels of evidence for each gene’s association with ASD. So far, 418 high-confidence and strong candidate ASD risk genes have been identified.

Figure 1 summarizes the systems level approach to selecting the targets from animal models and identifying those that are also listed in SFARI. Such targets are listed centered in each system’s box. We provide further details on the selected targets in Table 1 with the level of each gene’s association indicated in squared brackets. Scores denoted as S identify syndromic category, and lower numbers in the range from 1 to 6 mean higher confidence in the association, with 1 being the highest and 6 being the lowest.

**Table 1 biology-12-00914-t001:** Genes implicated in the signature of astrocytes and microglia exposed to chronic inflammation or stress.

Groups	Genes	Function	Effect of Inflammation	Effect of Stress
Glial cell phenotype	*TMEM119*	Transmembrane protein 119; identifies resident microglia (from blood-derived macrophages) [8,9]	Up	Up
	*TGFβ* **[4]**	Transforming growth factor beta-1; resident microglial biomarker [10,11]	First up, then down	Down
	*CD11b*	Resident microglial biomarker [9,10,11]	Up	Up
	*CD11c*	Activated microglia [10,11]	Up	Up
	*CD45* **[4]**	Protein Tyrosine Phosphatase Receptor Type C (PTPRC); Resident microglial biomarker [9,10,11]	Up	Up
	*Iba1*	Ionized calcium binding adaptor molecule 1: Non-specific microglial/macrophage biomarker [3,9,10,11,12]	Up	Up
	*CX3CR1* **[4]**	CX3C chemokine receptor 1; required for synaptic pruning during brain development [13]	Up	Up
	*BRD4* **[4]**	Bromodomain containing 4; polarizes microglia toward inflammatory phenotype [14]; involved in epigenetic memory [15]	Up	Up
	*SLC1A2* **[S]**	Astrocytic GLT-1 transporter required for neuron–astrocyte communication and astrocyte maturation [16]	Up	Down
Inflammation	*HMGB1*	Hypermobility group box protein 1, a pleiotropic signaling molecule in glia cells and neurons: a growth factor, a pro-inflammatory molecule [17]; implicated in ASD [18]	Up	Up
	*IL-10*	Key regulator of neuroimmune homeostasis via cross-talk of microglia and astrocytes [19]	Down or Up	Down
	*IL-6* **[5]**	Early inflammatory cytokine	Up	Up
	*NLRP3*	Inflammasome activated in ASD [20]	Up	Up
Stress	*CRH* **[5]**	Corticotropin-releasing hormone, key hormone linking chronic stress with anxiety [21,22]; has direct effects on microglia [23]	Up	Up
	*CRHR2* **[5]**	Corticotropin-releasing hormone receptor 2; [23]	Up or Down, locoregional	Up
	*HSD11B1* **[4]**	11β-Hydroxysteroid dehydrogenase type 1 [24]	Up	Down
	*POMC*	Proopiomelanocortin [25]	Down	Up
	*p-Akt*	Phosphorylated-Akt [26]	Up	Up
	*PI3K* **[S]**	PI3K/Akt signaling pathway; e.g., PIK3CA [26,27]	Up	Up
	*OGT* **[5]**	O-GlcNAc transferase; a placental biomarker of maternal stress exposure related to neurodevelopmental outcomes [28,29]	Up	Up
	*GAP43* **[5]**	Growth associated protein 43; in astrocytes, GAP43 mediates glial and neuronal plasticity during astrogliosis and attenuates microglial activation under LPS exposure [30]	Down	Down
	*SLC22A3* **[5]**	Solute carrier family 22 member 3; modulates anxiety and social interaction [31,32]	Up	Up
	*PLPPR4* **[5]**	Phospholipid Phosphatase Related 4; stress-related behaviors such as reduced resilience [33]	Up	Up
	*PRKCB* **[3]**	Protein Kinase C Beta; involved in stress-related behavior [34]	Up	Up
	*UCN3* **[5]**	Urocortin 3; binds specifically CRHR2 [23]	Down	Down
	*DLG4* **[5]**	Disks large homolog 4; modulates stress reactivity and anxiety [35,36]	Down	Down
Energy homeostasis	*Cx43*	Connexin 43 gap junction maintaining astrocytes’ homeostasis via metabolic cooperation [37]	Down	Down
	*AMPK*	Adenosine monophosphate kinase, intracellular energy sensor [38]	Up	Up
	*FBP*	Fructo-biphosphokinase: signature of second-hit memory of inflammation in fetal microglia [3]	Up	Up
	*mTOR* **[S]**	Mammalian target of the rapamycin signaling pathway [39,40]	Up	Up
Iron homeostasis	*HAMP*	Hepcidin, a regulator of iron homeostasis and inflammation; implicated, along with ferroportin and transferrin, in microglial response to endotoxin interfering with α7nAChR signaling [4]	Up	Up
	*SLC40A1*	Ferroportin	Up	Up
	*TFR2*	Transferrin receptor 2; involved in iron sequestration	Up	Up
	*TFRC*	Transferrin receptor protein 1; needed for iron sequestration	Down	Down
	*HMOX1*	Hemoxygenase 1, key enzyme of iron homeostasis; also serves as a signature of second-hit memory in fetal microglia and promoted by α7nAChR stimulation [3,41]	Up	Up
	*SLC25A39* **[4]**	Member of the SLC25 transporter or mitochondrial carrier family of proteins; required for normal heme biosynthesis	Up	Up
Complement pathway	*C1QA*	Aside from their traditional role in innate immunity [42], elements of the complement pathway are involved in neuronal–glial interactions; recognized as essential players in brain development, especially in synaptogenesis/synaptic pruning and predisposition for neurodegenerative diseases [7,43]; their microglial expression is also susceptible to LPS exposure in utero [4]	Up	Up
	*C1QB*		Up	Up
	*C43AR1*		Up	Up
	*CR2*		Up	Up
	*C4B* **[4]**		Up	Up
Epigenetic memory	*HDAC* **[S]** and *HAT* families	Histone acetylation/deacetylation enzymes: HDAC 1, 2, 3, 7 and 9 found in SSC database of ASD gene mutations.HDAC 1, 2, **4** and 6 involved in fetal microglial memory of LPS exposure; HDAC 1, 2, **4** and 7 increased with altered neuronal AChE signaling and increased anxiety behavior due to adult chronic stress exposure [3,4,25,27,44,45]	Up or Down	Up or Down

Hits in the SFARI database are in bold. ASD association score indicated in square brackets: S: syndromic; 1: high confidence; 2: strong candidate; 3: suggestive evidence; 4: minimal evidence; 5: hypothesized; 6: not supported. For details, see https://gene-archive.sfari.org/about-gene-scoring/ (accessed on 8 May 2023). “Syndromic” distinction contains mutations that are associated with a substantial degree of increased risk and consistently linked to additional characteristics not required for an ASD diagnosis.

**Figure 1 biology-12-00914-f001:**
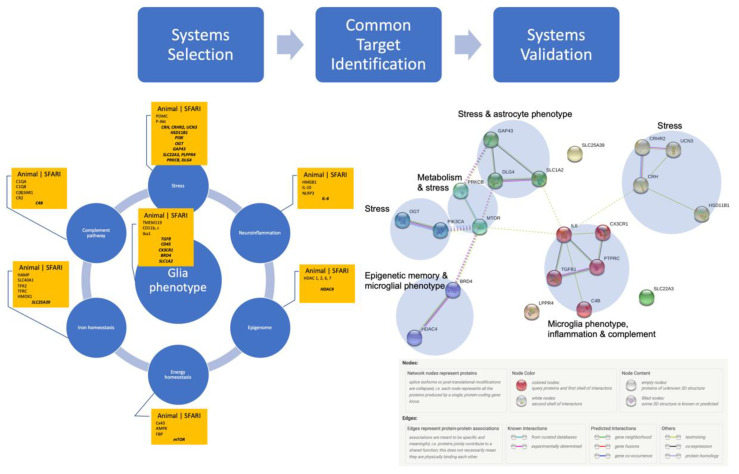
***** Systems-driven selection of potential targets from animal studies, followed by confirmation via SFARI database (centered in each box), thus identifying possible system targets reflecting ASD (see Table 1 for details). Systems validation using Homo sapiens genes followed, using the protein network analysis of ASD-associated genes driven by exposure to stress or inflammation followed by Markov Cluster Algorithm for unsupervised clustering identified significant interactions between six clusters containing microglial and astrocytic phenotypes, inflammation, stress, energy (but not iron) homeostasis, and complement pathway. Note that stress pathways are represented in four clusters (yellow, green, aquamarine, dark cyan), while microglial phenotype, inflammation and complement pathway form one cluster (red). Protein–protein interaction (PPI) enrichment p-value: 0.00085. PPI legend by string-db.org [46]. The detailed list of participating node members is provided in Appendix A [47]. * (Hi-res version at the end of the manuscript).

Based on the above-described etiological understanding of the developmental interplay between the glial cells, neurons and the following signaling pathways, the genes were grouped into those identifying the phenotypes of glial cells (microglia and astrocytes); and genes implicated in inflammation, stress, energy or iron homeostasis, complement pathway or epigenetic memory, while being implicated in ASD via preclinical or clinical studies, as referenced. Where applicable, we denote the effects of acute or chronic exposure to inflammation or stress on these genes. These notes are somewhat simplified for brevity; sometimes, a temporally biphasic or tissue-specific pattern was observed, and we made note of that, as appropriate.

We used the SFARI database (using Homo sapiens only) to identify genes with varying degrees of disruptive mutations associated with ASD, as defined in the SFARI database, to any of the genes in Table 1 (from any species). The analysis was based on accessing the database in August 2019 corresponding to the SFARI database update of 20 June 2019.

We identified 21 hits, indicated in Table 1 in bold along with the level of confidence for association with ASD using the SFARI scoring criteria. It was apparent that all categories defined in Table 1 were represented in the SFARI database: microglial or astrocytic phenotype, inflammation, stress, energy homeostasis, iron homeostasis, complement pathway, and epigenetic memory. The response to chronic inflammation or stress is a multifaceted one, depending on tissue type and duration/frequency of exposures. It is the system’s level pattern of changes that provides a clearer picture of the disease phenotype.

## 3. Results

### Systems Validation: Many Species, One Network

We sought to validate the significance of the association of the identified protein products shown in Table 1 using network analysis with string-db.org. The results can be accessed online at the permanent URL. They represent the analysis of the subset of 21 proteins identified in Table 1 against the background of all known protein–protein interactions in Homo sapiens. String-db extracted the data for this from Biocarta, BioCyc, Gene Ontology, KEGG and Reactome. In the version 11.0 we used, this corresponded to 24,584,628 proteins from 5090 organisms with a total of 3,123,056,667 interactions or 19,566 distinct protein-coding genes from Homo sapiens. While individually, 16 out of the 21 identified proteins were scored as weakly associated with ASD (scores 4 or 5, denoting minimal evidence or hypothesized), the network analysis showed a significant interaction between all 21 nodes on the protein–protein interaction level with six network clusters shown in Figure 1. These clusters were organized into subsets of networks associated with stress, metabolism, glia phenotype and epigenetic memory aspects of the proposed neuro-immunometabolic network exposed to stress or inflammation.

The data summarized in Figure 1 support our initial hypothesis linking both neuroinflammation and stress exposures in utero to the etiology of ASD in a gene-environment paradigm: developmental combinations of genetic alterations in key pathways susceptible to environmental stimuli of inflammation or stress increase risk for or cause ASD. The clusters of interactions map onto the neuro-immunometabolic and stress response networks.

## 4. Discussion

### 4.1. Information Processing, Energy Demand and Stress

Current estimates suggest that variation in as many as 1000 different genes could affect susceptibility to ASD. In an attempt to tie this wealth of data together conceptually in a unified model, a unified model of autism has been proposed [48]. It is defined as a predictive impairment.

Sinha et al. note in [48] that providing a cohesive conceptualization of ASD, such as the predictive impairment, would ameliorate the search for broadly effective therapies, diagnostic markers, and neural/genetic correlates. Indeed, setting the ASD on the information-theoretical platform in terms of prediction impairment reveals links to a major disruptor of neural information processing—the stress. For example, the prediction impairment hypothesis interprets the reduced habituation and hence greater stress in ASD subjects as being caused by an endogenous predictive impairment that leads the environmental stimuli to appear more chaotic. By the same token, an autistic individual responds to the chaotic-appearing environment with endogenous mechanisms geared to reducing the chaos by exhibiting ritualized behaviors. These features of the proposed ASD model as a predictive impairment all have in common an attempt to reduce stress arising from increased uncertainty.

Peters et al. provide a neurobiological perspective on these psychological observations [49]. Their observations are based on the well-founded physical-mathematical concepts linking the energy expenditure, entropy and information processing, going back to Shannon and Boltzmann [50,51]. This is combined with a body of neurobiological literature on the neuronal substrate of the brain’s stress and prediction networks [49]. Herein, the so-called Selfish Bayesian Brain resolves stress, or information uncertainty, by an internal Bayesian update of the knowledge state. This must take place at the expense of transiently increased energy needs (hence “selfish”). If this update succeeds, the stress is a eustress; if it does not, it becomes a distress. The present findings suggest that one possible mechanism for when such an update can fail is when the increased energy demands cannot be met.

### 4.2. Defining a Cohesive Cell Systems Correlate of ASD as Predictive Impairment

We are proposing a developmental origins perspective on this framework. The neuro-immunometabolic and stress response networks interact, and in utero exposure to stress or inflammation may alter the glial immunometabolic phenotype, i.e., the pattern of this network interaction. That consequence translates into less energy availability under the increased demands due to stress [49]. If the energy needs are not met, the allostatic load turns into an allostatic overload [52]. A neuropathophysiological state may ensue unless the situation is corrected.

The above findings and their interpretation are further supported by the emerging evidence for the dynamic relationship between epigenetic memory and energy availability. In a broad sense, epigenetics refers to the chemical mechanisms that modify gene expression, without altering the DNA sequence. The main epigenetic modifications that change the chromatin architecture include DNA methylation, histone post-translational modifications and micro-RNA-mediated repression. While epigenetic regulation is dynamic, it is often also referred to as epigenetic memory, implying the stable propagation of changes in gene expression induced by a developmental or environmental stimulus [53]. Environmental stimuli include nutrients and energy metabolites to which we are exposed regularly and that are known to regulate gene expression through modulation of the epigenome. This paradigm was recently termed “metaboloepigenetics”, bringing forward the crucial role of metabolites as co-factors of chromatin-modifying enzymes that are responsible for epigenetic modifications [54,55]. Enzymes such as methyltransferases, deacetylases and kinases employ cofactors such as α-ketoglutarate, acetyl-CoA, nicotinamide adenine dinucleotide, S-adenosylmethionine and ATP, and the availability of these cofactors as well as nutrients such as glucose, oxygen and glutamine can reduce or enhance gene expression [56]. This implies that the response to a stressful microenvironment, such as intrauterine or postnatal adversity, will ultimately lead to a pathological state. Exposure to in utero stress and/or inflammation is likely to result in impaired levels of micronutrients and cofactors, leading to a pathological methylation/demethylation of DNA and histone proteins and histone acetylation/deacetylation of the genes clustered in the neuro-immunometabolic network proposed. The presence of such effects can be validated in future preclinical and clinical studies.

## 5. Limitations and Future Directions

The proposed neuro-immunometabolic hypothesis can be used in future in vivo studies to probe causally for the role of the immunometabolic gene and protein networks in ASD etiology, in particular in the areas described as being the core of both the prediction processing network and contextual fear memory, the anterior cingulate cortex [57,58,59]. Indeed, a recent study reported that an immunometabolic dysregulation reduces the thickness of the anterior cingulate cortex, while another study revealed 1223 differentially methylated genes (5018 differentiated methylated regions, DMRs) in this brain region 4 weeks after contextual fear conditioning [60,61]. In another, most recent study in a human cohort of stressed pregnant mothers and their newborns, we identified novel associations between newborn epigenome-wide methylation status in saliva and chronic stress suffered by the mother during pregnancy [62]. Among the positively associated CpGs is the CpG annotated to *YAP1* (Yes1 regulated transcription factor) whose deletion has been related to reactive astrogliosis and astrocyte-driven microglial activation [63]. The protein of another gene (*CSMD1*, CUB And Sushi Multiple Domains 1) associated with maternal cortisol, CSMD1, seems to play a crucial role in regulating complement activation and inflammation in the developing brain [64,65]. Two DMRs were also found annotated to *DAXX* (Death-associated protein 6) and *ARL4D* (ADP-Ribosylation factor 4D), which together with *CSMD1* have all been directly or indirectly involved in neurodevelopmental disorders such as ASD, highlighting the impact of prenatal stress on the epigenetic landscape of the newborn in genes associated to neuroinflammation and ASD [66,67,68,69,70].

Moreover, inflammatory priming in maternal immune activation (MIA) can, in general, substantially alter neurodevelopment [71]. Specifically, microglia, as the key players in MIA, can modify neurogenesis, maturation of synapses and neural circuits. Due to various triggers, bacterial/viral or autoimmune triggers, as reviewed extensively elsewhere [71], MIA can influence the efficacy of microglia to sculpt neuronal connections. This can result in a dysfunctional neurodevelopment, culminating in ASD and other neurodevelopmental disorders in the offspring, such as attention-deficit/hyperactivity disorder and Tourette syndrome. Hence, inclusion of MIA studies might lead to additional insights into the role of neuro-immunometabolic changes in neurodevelopmental disorders such as ASD.

The therapeutic approaches may include prenatal and early neonatal screening using early biomarkers of altered brain development due to intrauterine exposure to stress or inflammation, followed by early postnatal reprogramming of these effects using behavioral (e.g., environmental enrichment) or bioelectronic tools to correct brain developmental trajectories at an early stage [2,72,73,74].

In the context of the discussed neuro-immunometabolic and Selfish Bayesian Brain paradigms, environmental enrichment may help reduce the allostatic load by providing opportunities to perform the required knowledge state update in the face of existing uncertainties, i.e., stress.

Other salutary mechanisms mediating the therapeutic approaches may involve activation of the efferent and afferent cholinergic pathways. The efferent wiring of the vagal cholinergic pathway has begun to unravel in the last decade, while our understanding of the afferent signaling is only beginning to emerge.

The efferent component is also referred to as the inflammatory reflex, which acts via the cholinergic anti-inflammatory pathway (CAP), a neural mechanism that influences the magnitude of innate immune responses to inflammatoQry stimuli and maintains homeostasis [75]. Through CAP, increased vagal activity inhibits the release of pro-inflammatory cytokines. We found that spontaneous CAP activity is present in the sheep fetus at as early as 0.76 gestation [76]. Moreover, CAP activation near-term suppresses activation of ovine microglia and astrocytes that express α7 nAChR or hypoxic-ischemic brain damage in newborn rats [12,17,77,78,79,80].

Vagal afferent signaling affects widely distributed areas of the brain via Nucleus tractus solitarii, and studies indicate that stimulation of the vagus nerve activates many neurons in both cerebral hemispheres and the hippocampus [81,82]. Vagal signaling in near-term fetus decreases HMGB1 concentrations in neuronal and astrocyte cytoplasm expressing α7nAChR and decreases microglial activation, which may decrease glial priming [12]. Hence, efferent peripheral or afferent vagal activity may mediate central neuroprotection via regulation of neuroinflammation. Vagus nerve stimulation has been shown to alter the phase synchrony in the anterior cingulate cortex, improving decision making in rats [83]. Several models of interactions between afferent vagus nerve signaling, stress, inflammation and glial plasticity as well as the risk for neurodevelopmental disorders have been proposed. Notably, the present analysis confirmed the implicated effects of stress and inflammation on iron homeostasis, yielding SLC25A39 as a target associated with ASD (cf. Figure 8 in [4]).

In the following, we indicate this study’s limitations and propose some solutions to overcome these in future work. Our model and hypothesis presented in this study may be further strengthened and deepened through validation via a model-free approach. This can be achieved by analyzing the individual genomic and transcriptomic information from the same SFARI database and evaluating the evidence of the presence of the observed networks—by investigating independent datasets that were not utilized in deriving the networks in Figure 1 by means of methodologically distinct analytic approaches. For this purpose, we envision the use of the following three computational pathway and network analysis schemes. First, we may quantitatively assess the relevance and relative importance of known pathways (e.g., canonical pathways curated in KEGG database) based on the differential activity of each pathway under different conditions (e.g., whether or not an individual has experienced exposure to fetal neuroinflammation and prenatal stress). It has been shown that differential pathway activity can be effectively inferred from gene expression data using probabilistic graphical models, and that highly differentially activated pathways detected therefrom can serve as better biomarkers for complex diseases (such as cancer) with higher discriminative power as well as improved reproducibility [84]. Next, we may identify differentially expressed genes or differentially activated pathways as “seeds” and identify potential network modules that include these seeds. For example, we may adapt the strategy proposed by Wang et al. [85], in order to find out whether the seed genes/pathways would be part of a well-conserved network module with characteristic topological features often sighted in known functional modules (e.g., signaling pathways, protein complexes). Another promising approach is to carry out a network-based analysis of the transcriptomic data in integration with protein interaction data for unsupervised detection of potential network modules that may be associated with the cellular response to fetal neuroinflammation and prenatal stress. As shown in [86], recent advances in deep learning (esp., graph convolutional networks) can facilitate such investigation. These remain the subjects of our ongoing research.

## 6. Conclusions

The immediate take-away from the present work is the putative common foundation of neuro-immunometabolic networks, on genes and protein levels, as etiology of ASD.

In context with the literature, from a translational viewpoint, we speculate that early behavioral therapy or enhancing fetal CAP activity via vagal nerve stimulation or α7nAChR agonists will suppress the activation of microglia and astrocytes, restoring their physiological immunometabolic phenotype, thus decreasing glial priming and preventing a sustained switch to a reactive phenotype. This approach might decrease the degree of brain injury sustained from fetal systemic and brain inflammatory responses and stress exposures, thus improving postnatal short- and long-term health outcomes, including susceptibility to ASD.

## Data Availability

All data are available on sfari.org and https://github.com/martinfrasch/ASD_origins_hypothesis (accessed on 8 May 2023). The data are citable as [47] and accessible on GitHub: https://github.com/martinfrasch/ASD_origins_hypothesis (accessed on 8 May 2023).

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
