# Peer review of "Autism Spectrum Disorder: A Neuro-Immunometabolic Hypothesis of the Developmental Origins"

_biology, 2023, doi:10.3390/biology12070914_

Round 1

Reviewer 1 Report

This manuscript proposed a neuro-immunometabolic etiological model of autism spectrum disorder (ASD) that integrated the data across the scales of physiological organization, from genome to phenotype. The subject is very interesting and the manuscript is well structured. I have the following suggestions:

1. Although the mechanism of ASD remains elusive, the authors need to present the current status of the ASD mechanism in the Introduction.

2. Figure 1 does not present quality and is difficult to read.

3. Methods fail to show how the authors performed the gene identification.

4. Results fail to show the connections between fetal neuroinflammation, PS and ASD, especially based on the current data. 

5. Synthesis and Discussion can be combined.

6. Conclusions should be based on the existing results and the current conclusions extended too widely.

Author Response

  1. Although the mechanism of ASD remains elusive, the authors need to present the current status of the ASD mechanism in the Introduction.

Response: We added some background information as the reviewer suggested. Thank you.

  1. Figure 1 does not present quality and is difficult to read.

Response: We apologize. We added the figure in better quality. There is also a hi-res version of it available for download. This is also indicated with * in Figure 1 legend.

Of note, the figure can be magnified in PDF viewer so all text details become legible.

  1. Methods fail to show how the authors performed the gene identification.

Response: We revised accordingly. Lines 103-119 and Lines 124-132 report on how the target genes were selected. This is summarized in Table 1 with corresponding references. We revised Table 1 for more context on the genes selected (showing how chronic exposure to inflammation or stress affects them).

  1. Results fail to show the connections between fetal neuroinflammation, PS and ASD, especially based on the current data. 

Response: The gene selection was based on their - experimentally established - role in fetal neuroinflammation, PS and ASD. This is explained in Methods where we present how the Table 1 was constructed. We added more detail now on Lines 113-119.

The connections between the genes arise from the statistical analysis of the corresponding networks which makes it unlikely that such connections exist by mere chance. This is reported on Lines 144-154 and in Figure 1 (Lines 156-165)..

  1. Synthesis and Discussion can be combined.

Response: Thank you for this advice. We have done so.

  1. Conclusions should be based on the existing results and the current conclusions extended too widely.

Response: We appreciate this comment and agree. We revised the conclusions accordingly. 

Reviewer 2 Report

Reviewer Comments:

The manuscript titled “autism spectrum disorder: a neuro-immunometabolic hypothesis of the developmental origins" by Martin G. Frasch et al shows a comprehensive conclusion and report the significant association of various genes involvement belongs to the glial cells, inflammation, Stress, Energy & iron homeostasis and complementary pathways, which are linked to the etiology of autism spectrum disorder (ASD). The manuscript has great merits but encountered minor problems.

1.     Authors need to explain the size of gene database of   Simons Foundation Autism Research Initiative (SFARI) Gene database.

2.     The authors have mentioned various genes and hormones in Table 1-Are these genes got mutated and the hormone levels got upregulated/downregulated-need to mention.

3.     Authors need to explain in detail in the proposed model about the signaling pathway-how the CAP gets activated via vagus nerve stimulation which in turn how it suppresses the activation of microglia and astrocytes and ultimately leads to degree of brain injury and susceptibility to ASD.

4.      English language grammar editing is mandatory throughout the manuscript before accept for publication.

Author Response

  1.     Authors need to explain the size of gene database of   Simons Foundation Autism Research Initiative (SFARI) Gene database.

Response: Thank you for this advice. We added this information (Lines 103-107).

  1.     The authors have mentioned various genes and hormones in Table 1-Are these genes got mutated and the hormone levels got upregulated/downregulated-need to mention.

Response: Thank you for this comment. We agree that the original presentation lacked detail about the changes found in the listed genes, albeit we did refer to the original publications.

We now revised the presentation including what changes were observed in the listed genes in relation to exposures to chronic inflammation or stress (Table 1 revised and Lines 116-119 & 130-132).

The response to inflammation or stress exposures is a multifaceted one, depending on tissue type and duration/frequency of exposures. It is the system’s level pattern of changes that provides a clearer picture of the disease phenotype. That is what we are hoping to get across as the key finding in the present manuscript.

  1.     Authors need to explain in detail in the proposed model about the signaling pathway-how the CAP gets activated via vagus nerve stimulation which in turn how it suppresses the activation of microglia and astrocytes and ultimately leads to degree of brain injury and susceptibility to ASD.

Response: Thank you for this advice. We added this information (Lines 249-267).

  1.     English language grammar editing is mandatory throughout the manuscript before accept for publication.

Response: Thank you. We carefully reviewed the manuscript in that regard.

Reviewer 3 Report

The authors argue that the seed of neurological disorders begins during the fetal development due to the prenatal stress and neuroinflammation specifically astrocytes & microglia. They propose a hypothesis, a new model of neuro-immunometabolism on the origins of ASD. The authors analyze the transcriptome data from ovine & rodent exposed to neuroinflammation or PS retrieved from Simons Foundation Autism Research Initiative Gene Database. The authors claim to confirm 21 genes & identify 6 clusters of probable PPIs..

The manuscript could have been of value to the community, if the model could have been described better. However, in the current state, the manuscript is hard to read and fails to convey their results, and consequently, the significance of their work. There should be mechanistic figures describing what the results are & what is the model.

The manuscript is hard to read. The results should be described in more detail.

Author Response

The manuscript could have been of value to the community, if the model could have been described better. However, in the current state, the manuscript is hard to read and fails to convey their results, and consequently, the significance of their work. There should be mechanistic figures describing what the results are & what is the model.

Response: Thank you for your comments. 

You can see our responses to Reviewers 1 and 2 which also noted deficiencies in the explanation of methodology and results. Please find enclosed a major revision of the manuscript. We hope this version addresses this reviewer’s concerns. 

Comments on the Quality of English Language

The manuscript is hard to read. The results should be described in more detail.

Response: Thank you. We carefully reviewed the manuscript in that regard.

Specifically, we expanded the methodological approach as well as the results in context with the string-db.

Round 2

Reviewer 1 Report

The manuscript can be accepted in its present form.